# NoCoLA: The Norwegian Corpus of Linguistic Acceptability

**Matias Jentoft** and **David Samuel**
University of Oslo, Language Technology Group
{matiasj, davisamu}@ifi.uio.no

## Abstract

While there has been a surge of large language models for Norwegian in recent years, we lack any tool to evaluate their understanding of grammaticality. We present two new Norwegian datasets for this task. **NoCoLA**$_{class}$ is a supervised binary classification task where the goal is to discriminate between acceptable and non-acceptable sentences. On the other hand, **NoCoLA**$_{zero}$ is a purely diagnostic task for evaluating the grammatical judgement of a language model in a completely zero-shot manner, i.e. without any further training. In this paper, we describe both datasets in detail, show how to use them for different flavors of language models, and conduct a comparative study of the existing Norwegian language models.

## 1 Introduction

Large pre-trained language models have recently led to a revolution in natural language processing (NLP) as they substantially increased the performance of most NLP tools (Peters et al., 2018; Devlin et al., 2019). Large language models were originally developed for English, but a surge of Norwegian-based models has recently followed (Kutuzov et al., 2021; Kummervold et al., 2021; Hofmann et al., 2022). The remaining issue is that the Norwegian linguistic resources do not contain a large range of tasks to evaluate and compare these models on, as opposed to the English benchmark suites like GLUE (Wang et al., 2018), SuperGLUE (Wang et al., 2019) or GLGE (Liu et al., 2021), to name a few.

We present two new datasets for evaluating the understanding large language models have of Norwegian grammar, jointly called the Norwegian corpus of linguistic acceptability (NoCoLA). We hope

```
# Incorrect (inflection):
    Samfunnet ville bli mer fornøyet.
# Correct:
    Samfunnet ville bli mer fornøyd.

# Incorrect (word choice):
    Jeg er ikke nordmann, med jeg trives i Norge.
# Correct:
    Jeg er ikke nordmann, men jeg trives i Norge.
```

Listing 1: Two illustrative examples of incorrect / correct sentence pairs from **NoCoLA**$_{zero}$. The English translations: *"Society would be happier"* and *"I'm not Norwegian, but I enjoy living in Norway."*

that the datasets can contribute to the development of a language model benchmark suite for Norwegian. Our work is limited to the most widely used of the written standards for Norwegian, namely Bokmål. This paper proposes two different views on the same set of sentences, each with a slightly different purpose:

- **NoCoLA**$_{class}$ is a collection of sentences split into two classes: grammatically acceptable and non-acceptable. Thus, it is a binary classification task, where a language model is expected to be first fine-tuned on the training data split. This task is more practically-oriented and evaluates the fine-tuning abilities of a language model. The downside is that we cannot tell if the performance comes from its innate abilities or if it was obtained from the supervised fine-tuning.

- **NoCoLA**$_{zero}$ is a collection of pairs of sentences, where only one of them is grammatically acceptable. Here, we do not fine-tune on this task at all, the language model gives a probability to each of the two sentences, and we measure how often the correct one gets a higher probability. While not as practical as the first task, the zero-shot evaluation provides a better estimate of the innate grammatical understanding.

We provide a comprehensive evaluation of the existing Norwegian language models and release the

data and code for an easy evaluation of new Norwegian models.[1]

## 2 Related work

The closest equivalents of our **NoCoLA**$_{class}$ dataset are the English Corpus of Linguistic Acceptability (CoLA; Warstadt et al., 2019) and the Swedish Dataset for Linguistic Acceptability Judgments (DaLAJ; Volodina et al., 2021). On the other hand, **NoCoLA**$_{zero}$ roughly follows the The Benchmark of Linguistic Minimal Pairs for English and the English (BLiMP; Warstadt et al., 2020).

**Data sources.** There are two primary strategies for obtaining non-acceptable sentences for a corpus of linguistic acceptability. The non-acceptable sentences are either collected from the linguistics literature by experts as in the English, Russian and Italian corpora (Warstadt et al., 2019; Mikhailov et al., 2022; Trotta et al., 2021) – or these sentences are collected from natural texts, usually based on the language of language learners, such as in the Swedish acceptability corpus (Volodina et al., 2021). The second, natural, approach is also used for the creation of our Norwegian corpus.

**CoLA.** This dataset consists of 10 600 acceptable and non-acceptable sentences collected manually from the linguistics literature, with the goal of covering specific linguistic phenomena – and the morphological, syntactic and semantic violation of rules connected to those phenomena. By collecting the data in this manner, one ensures that the dataset represents language phenomena that are central to human linguistic competence according to linguists. CoLA has become a standard task for evaluating English language models after it was included in the GLUE benchmark for natural language understanding (Wang et al., 2018). Similar datasets for Russian (RuCoLa; Mikhailov et al., 2022) and Italian (ItaCoLA; Trotta et al., 2021) follow the same methodology as the English CoLA.

**DaLAJ.** This dataset has the same purpose for benchmarking Swedish language models as CoLA has for English. In contrast to the English CoLA, DaLAJ uses the error-annotated learner corpus SweLL (Volodina et al., 2019) as their source of non-acceptable sentences. DaLAJ contains 4 798 sentence pairs, where the non-acceptable versions are annotated with one of four error-tags. All of

the error-types in DaLAJ vol.1 focus on semantic aspects of the sentences, and morphological and syntactic error types are left for future work. The original sentences are edited so that each sentence only has one error focus.

**BLiMP.** The BLiMP dataset consists of 67 000 minimal pairs, all of them generated artificially. Some examples of phenomena covered in the dataset are determiner-noun agreement, verb argument structure and irregular verb-forms. Each pair differs only on one single parameter, namely the element that leads to the non-acceptability.

**Comparison with NoCoLA.** Our datasets fill the same purpose for evaluation of language models in Norwegian as CoLA and BLiMP do for English. However, the source of the sentences is different, as we follow the methodology used for DaLAJ. Our data consists of naturally produced sentences, instead of controlled and artificially generated ones. Where CoLA collects sentences that are handpicked by linguists to represent specific linguistic phenomena, our sentences contain errors that mirror the natural distribution of errors in texts by second language learners. Thus, NoCoLA gives an indication of how well a given language model distinguishes between acceptable and non-acceptable Norwegian text, but not of how well it understands the full range of possible grammatical phenomena of the language. NoCoLA is also substantially larger than CoLA, with almost 15 times more examples. The NoCoLA error types are not comparable to BLiMP, where the error-types describe the underlying grammatical problem. Instead, the NoCoLA error-types describe the changes that need to be made to correct the errors. In contrast to DaLAJ we keep original sentences belonging to all error type categories, including morphological, syntactic, and semantic errors.

## 3 Datasets description

### 3.1 ASK corpus

Both **NoCoLA**$_{class}$ and **NoCoLA**$_{zero}$ require a source for both acceptable and non-acceptable sentences. The latter is hard to come by in most naturalistic text by adult native speakers. Our source for both NoCoLA datasets is the ASK Corpus – A Language Learner Corpus of Norwegian as a Second Language (Tenfjord et al., 2006). It consists of submissions by second language learners of Norwegian Bokmål around the year 2000, each

---

[1] https://github.com/ltgoslo/nocola

with one or more essays. The essays are written as solutions to two separate Norwegian language exams, which are estimated in Berggren (2019) to be approximately CEFR-levels B1 and B2.

There are 1 935 submissions, with 46 000 original sentences in total. Each essay has been manually corrected by native speakers, hereby called correctors. The errors in the corpus are annotated with a set of error-codes, which indicate the change that needs to be done to correct the original passage. For instance, "F" indicates wrong morpho-syntactic category, while "PUNCM" means that punctuation is missing, and needs to be added. We have merged some of the error-codes so that we have a medium-grained way of understanding the performance of the models on the different types of errors found in **NoCoLA**$_{zero}$. A short explanation of these error-codes can be found in the appendix.

### 3.2 Conversion from ASK to NoCoLA

**Privacy considerations** The original ASK corpus is annotated with rich metadata about the learners. For this dataset we have decided to surpass all this metadata, including the CEFR-level of the test. ASK has also gone through a anonymization process, where possibly sensitive words have been replaced by placeholders. Still, some of the topics of the essays deal with so specific topics about the lives of the learners, that we decided to sentence-scramble the essays to achieve maximum anonymity.

**Sentence merging.** For the NoCoLA datasets we want sentences as the unit for evaluation. Therefore we need to split the continuous text of ASK into sentences. However, since some of the corrections suggested by the correctors affect the way the text is split into sentences, and we need alignment between the acceptable and non-acceptable in the pairs for **NoCoLA**$_{zero}$, we decided to always keep the longest available version in cases where there is disagreement between both versions. The principle applies to both datasets. Thus, the unit referred to as "sentence" in this paper can consist of multiple sentences.

**Error extraction.** For each of these sentences, we first extract a corrected (acceptable) version. In order to test only minimal errors and to label each non-acceptable sentence with an error-type, we generate one non-acceptable sentence for each error found in the originals. Therefore we extract

| Dataset | Train | Dev | Test |
|---|---|---|---|
| **NoCoLA**$_{class}$ | 116 195 | 14 289 | 14 383 |
| **NoCoLA**$_{zero}$ | — | — | 99 115 |

Table 1: Number of sentences and sentence pairs, respectively, for both NoCoLA datasets.

almost 100 000 non-acceptable sentences, as many of the original sentences have multiple errors.

**Post-processing.** We did a few additional adjustments to the dataset. All sentences are heuristically detokenized and removed if they contain an uneven count of quotation marks. If no error type is mentioned for a given correction, we also remove that sentence. The sensitive words that have been replaced by placeholders like "@sted" (place) and "@navn" (name) are replaced with a substitute representation of that category, i.e. "Oslo" instead of "@sted", to normalize all sentences. This is to avoid feeding too many unknown tokens to the language models. In rare occasions, these replacements might cause some sentences to become erroneous, since the possible genitive and plural conjugations in the original texts are not annotated with separate placeholder-tokens.

**Conversion results.** The final dataset contains 144 867 sentences, 31.5% of which are acceptable. **NoCoLA**$_{class}$ has been shuffled and then randomly split to ensure unbiased development and test sentences. The split has been done in an approximate 80:10:10 ratio, resulting in the sentence-level statistics from Table 1.

## 4 Baseline models

### 4.1 Evaluation of NoCoLA$_{class}$

In order to evaluate language models on **NoCoLA**$_{class}$, we use the standard fine-tuning approach from Devlin et al. (2019). Accordingly, every sentence is tokenized, prepended by a special [CLS] token, appended by a [SEP] token and input to a pre-trained language model. Subsequently, the contextualized representation of the special [CLS] token is fed into a binary MLP classifier. The pre-trained weights of the language model are further trained together with the classifier weights.

| Model | Inflection | Word choice | Spelling | Missing | Superfluous | Punctuation | Word order | Capitalization | Compounding | Derivation | Overall |
|---|---|---|---|---|---|---|---|---|---|---|---|
| BERT$_{base}$ (Devlin et al., 2019) | 50.70 | 53.55 | 63.43 | 60.44 | 51.69 | 79.33 | 51.85 | 82.54 | 54.31 | 54.11 | 59.48 |
| mBERT$_{base}$ (Devlin et al., 2019) | 79.92 | 69.05 | 90.74 | 76.91 | 78.84 | 83.97 | 74.88 | 87.88 | 78.72 | 80.44 | 79.53 |
| XLM-R$_{base}$ (Conneau et al., 2020) | 91.43 | 85.28 | 92.60 | 87.43 | 87.56 | 83.93 | 84.33 | 90.60 | 89.63 | 91.96 | 88.02 |
| ScandiBERT (Hofmann et al., 2022) | 93.43 | 89.79 | 90.84 | 90.14 | 90.05 | 87.10 | 90.08 | 90.55 | 85.82 | 90.68 | 90.27 |
| NB-BERT$_{base}$ (Kummervold et al., 2021) | 93.76 | 89.19 | **97.14** | 86.54 | **92.48** | 73.98 | 90.94 | 92.73 | **91.15** | **94.70** | 89.04 |
| NorBERT$_1$ (Kutuzov et al., 2021) | 93.46 | 88.46 | 94.54 | 88.66 | 89.41 | **88.46** | 92.01 | **94.26** | 90.83 | 93.05 | **90.83** |
| NorBERT$_2$ (Kutuzov et al., 2021) | 91.66 | 88.20 | 96.88 | 89.22 | 90.91 | 75.82 | 92.67 | 93.13 | 74.18 | 92.69 | 88.51 |
| NorBERT$_{3,\ base}$ (Samuel et al., 2023) | 94.63 | 90.98 | 87.06 | 91.04 | 90.23 | 87.25 | 89.82 | 92.73 | 86.95 | 89.21 | 90.44 |
| XLM-R$_{large}$ (Conneau et al., 2020) | 92.54 | 88.17 | 90.06 | 88.57 | 89.28 | 80.84 | 84.52 | 91.35 | 89.70 | 93.24 | 88.27 |
| NB-BERT$_{large}$ (Kummervold et al., 2021) | **95.20** | **92.41** | 95.16 | **91.47** | 91.92 | 85.33 | **93.36** | 17.01 | 89.56 | 92.87 | 90.51 |
| NorBERT$_{3,\ large}$ (Samuel et al., 2023) | 94.89 | 91.98 | 83.71 | **91.47** | 90.84 | 86.39 | 87.87 | 92.48 | 84.19 | 88.30 | 90.01 |

Table 2: The accuracy values of zero-shot evaluation on **NoCoLA**$_{zero}$. Fine-grained results over different error types are reported (Appendix A), as well as the overall average over all sentence pairs in the datasets.

## 4.2 Evaluation of NoCoLA$_{zero}$

One disadvantage of **NoCoLA**$_{class}$ is that the results are skewed by the second-stage supervised training and it can be problematic to disentangle the properties of the LM from the classifier (Belinkov, 2022). In contrast, pure LM-based evaluation of **NoCoLA**$_{zero}$ attempts to measure the linguistic knowledge of a language model in a zero-shot manner – without any additional training. The dataset consists of 99 115 sentence pairs; each pair differs minimally on the surface level, but only one of the sentences is acceptable. We can use the intrinsic ability of language models to assign a probability to every sentence and test how often a language model assigns a higher probability to the correct sentence, as in (Warstadt et al., 2020). As the two classes are perfectly balanced, simple accuracy is a sufficient metric for this setup.

**CLM evaluation.** The *causal* language models are trained to estimate $p(s_t|s_{<t})$ for sentence $s$ and token $s_t$ where $s_{<t} = (s_i|i < t)$; then the sentence log-probability is simply given by $\log p(s) = \sum_{t=1}^{N} \log p(s_t|s_{<t})$.

**MLM evaluation.** The issue with *masked* language models is that they are not designed to calculate the joint probability; they are trained to estimate $p(s_t|s_{\setminus t})$ – the likelihood of a token $s_t$ given its bidirectional context $s_{\setminus t} = (s_i|i \neq t)$. We can however still use MLMs to infer a *score* for each

| Model | Lang. | Size | Accuracy | MCC |
|---|---|---|---|---|
| BERT$_{base}$ | en | 110M | 69.56$^{\pm0.37}$ | 23.99$^{\pm0.41}$ |
| mBERT$_{base}$ | multi | 178M | 75.28$^{\pm0.66}$ | 46.39$^{\pm0.67}$ |
| XLM-R$_{base}$ | multi | 278M | 79.29$^{\pm0.20}$ | 55.14$^{\pm0.36}$ |
| ScandiBERT | multi | 124M | 80.25$^{\pm0.33}$ | 57.12$^{\pm0.37}$ |
| NB-BERT$_{base}$ | no | 178M | 80.69$^{\pm0.44}$ | 58.10$^{\pm0.48}$ |
| NorBERT 1 | no | 111M | 71.53$^{\pm0.80}$ | 35.85$^{\pm1.70}$ |
| NorBERT 2 | no | 125M | 79.99$^{\pm0.27}$ | 56.09$^{\pm0.30}$ |
| NorBERT$_{3,\ base}$ | no | 123M | 81.50$^{\pm0.16}$ | 59.21$^{\pm0.28}$ |
| XLM-R$_{large}$ | multi | 560M | 81.03$^{\pm0.27}$ | 58.56$^{\pm0.30}$ |
| NB-BERT$_{large}$ | no | 355M | 81.43$^{\pm0.32}$ | 59.68$^{\pm0.14}$ |
| NorBERT$_{3,\ large}$ | no | 353M | **82.48**$^{\pm0.21}$ | **60.96**$^{\pm0.45}$ |

Table 3: Accuracy and the Matthews correlation coefficient (Matthews, 1975), the main metric of **NoCoLA**$_{class}$. We report the mean and standard deviation across five runs on the test split.

sentence where a higher *score* corresponds to a more likely sentence. Wang and Cho (2019) defined *pseudo-log-likelihood score* of a sentence $s$ with model $\theta$ as

$$\text{PLL}(s) = \frac{1}{N} \sum_{t=1}^{N} \log p(s_t|s_{\setminus t}; \theta).$$

Salazar et al. (2020) tested PLL and found that it produces accurate predictions on BLiMP. We adopt

their approach and evaluate our models with PLL.

## 5 Results

### 5.1 Results on NoCoLA$_{class}$

The results from benchmarking the publicly available Norwegian language models on the classification task can be seen in Table 3. The classification accuracy is around 80% for for these models. One exception is the slightly older NorBERT 1, which performs substantially worse, even if being trained on clean Norwegian data: Wikipedia and newspaper articles (Kutuzov et al., 2021). We use the English BERT$_{base}$ as a naive baseline, which gives us a lower bound on the performance of any decent Norwegian language models. The three largest models give a small increase in performance compared to the base-sized versions of the same models. The NorBERT$_3$ models (Samuel et al., 2023) consistently outperform other models on this task.

### 5.2 Results on NoCoLA$_{zero}$

On the raw zero-shot diagnostic task (Table 2), all models trained on Norwegian or Scandinavian languages perform well with results around 90% accuracy. The best performance comes, perhaps surprisingly, from NorBERT 1 – possibly because it was pre-trained on a relatively small and clean corpus. Remarkably, increased number of parameters does not seem to improve performance on this task.

We have also included accuracy scores for the individual error-types, as these fine-grained scores can be used as a helpful cue for NLP researchers who develop new language models. Comparably low scores can signal a problem with their training corpus or with their tokenizer. For example, the two NB-BERT models are relatively weak on punctuation-related errors. The large version is trained on uncased data, which explains this models inability to understand the case-related errors. ScandiBERT performs comparably to the Norwegian ones on most parameters except for spelling.

## 6 Conclusion

In this paper we have proposed NoCoLA, the first dataset for linguistic acceptance in Norwegian Bokmål. We showed how to use it for measuring the linguistic knowledge of language models on both a classification task and a zero-shot probability comparison task. We have described how the datasets were created and what their motivation is, compared them to related work in English NLP

and showed how to use them for fine-grained error analysis of language models.

Lastly, we evaluated existing Norwegian masked language models on both proposed tasks. These results suggest that models trained specifically for Norwegian or Scandinavian languages perform better at discriminating between acceptable an nonacceptable sentences. The classification results also show that linguistic acceptability is a relatively hard task, as only one of the models achieved more than 60% on the main MCC metric. The results on our diagnostic dataset highlight some shortcoming of the existing models. We will release all evaluation sources in the camera-ready version.

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

## A    NoCoLA*zero* error types[2]

- **Inflection:** wrong form of word. Merged from ASK-codes "`F`": wrong morpho-syntactic form and "`INFL`": suffix from correct category, but wrong form for this particular word. *"Jeg vet ikke hvorfor jeg har valgt **dette** oppgaven." "I do not know why I have chosen this task."*

- **Word choice:** wrong choice of word. Merged from ASK-codes "`W`": wrong word and "`FL`": word from another language. *"Jeg er et eksempel **for** det." "I am an example of that"*

- **Spelling:** wrong spelling of word, corresponding to ASK-code "`ORT`". *"De er en rik **fammilie**." "They are a rich family."*

- **Missing:** word should be added. Corresponding to ASK-code "`M`". *"Norge kan bidra veldig mye på Europeiske planet." "Norway can contribute a lot at **the** European level."*

- **Superfluous:** word should be removed. Corresponding to ASK-code "`R`". *"Da mistet jeg den beste vennen **min** i hele livet mitt." "Then I lost the best friend in my whole life."*

- **Punctuation:** add or remove punctuation. Corresponding to ASK-codes "`PUNC`", "`PUNCM`" and "`PUNCR`". *"Hva skal jeg gjøre etterpå**.**" "What should we do afterwards?"*

- **Word order:** wrong order of words or phrases. Corresponding to ASK-code "`O`". *"Hvis du har tillatelse, **du kan** fiske også." "If you have a licence, you can fish as well."*

- **Capitalization:** add/remove capitalization. Corresponding to ASK-code "`CAP`". *"**nå** liker jeg meg godt i Oslo." "Now I enjoy myself in Oslo"*

- **Compounding:** deviation regarding compounding. Corresponding to ASK-codes "`PART`" and "`SPL`". *"**Etter på** skal jeg studere for å bli sykepleier." "Afterwards I want to study to become a nurse."*

- **Derivation:** deviation regarding derivation. Corresponding to ASK-code "`DER`". *"Derfor er jeg helt enig med **forbudelse** mot krenkende uttalelser." "Therefore I completely agree with the ban on offensive statements."*

- **Other:** any other error.

---

[2]The codebook given to the correctors of ASK contains a multitude of additional tags which are not used by the correctors. For example there is OINV for "subject/verb inversion i contexts where there should not be one".

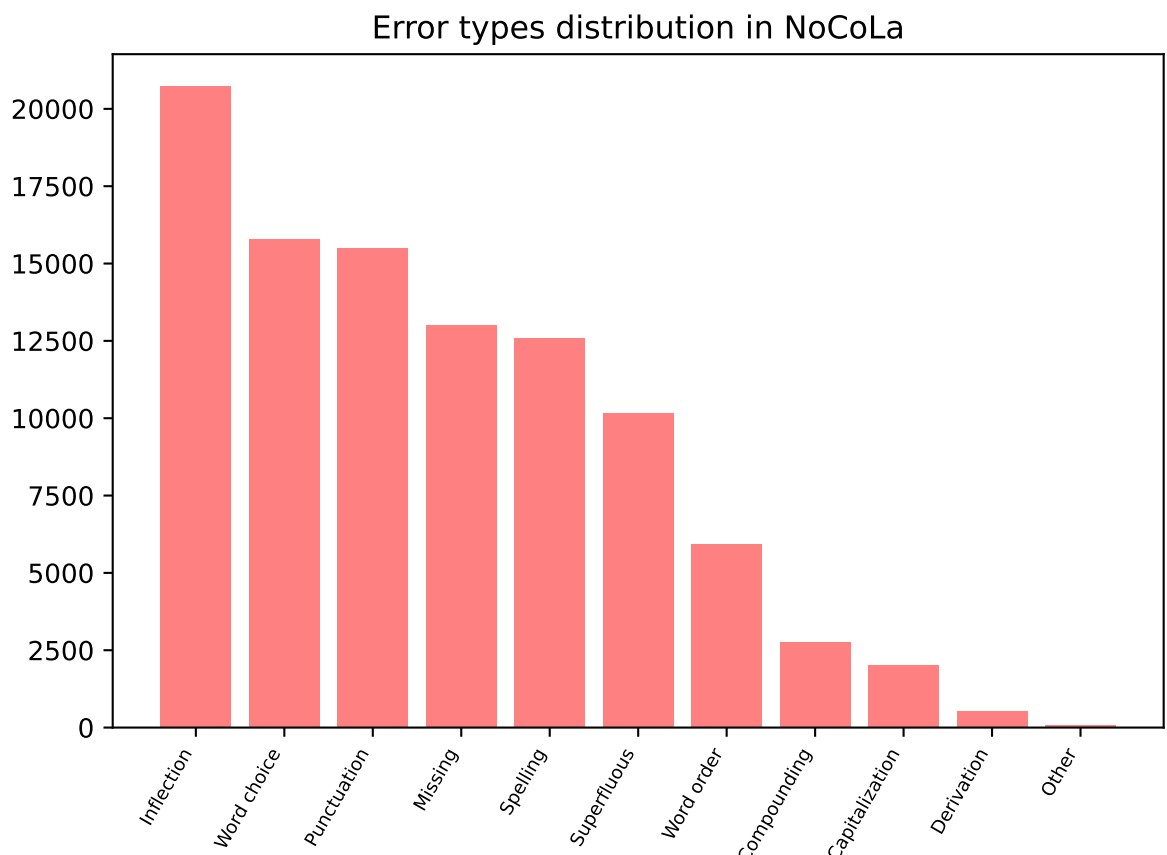

Figure 1: Distribution of error types in the NoCoLA datasets.