# OpenReview forum: "NoCoLA: The Norwegian Corpus of Linguistic Acceptability"
_NoDaLiDa/2023/Conference — NoDaLiDa 2023_

### Official Review · Reviewer_7noS · 2023-03-05
**Good dataset, limited context**

**Rating:** 6
**Confidence:** 4

**Review:**

The dataset presented is useful, and it's applaudable that other languages than English establish benchmarks for recognized NLP tasks. However, the paper provides way too limited context of previous work on similar datasets and on the area of linguistic acceptability, and especially on corresponding GLUE / SuperGLUE benchmarks for other languages than English. It seems that at least to acknowledge such efforts, especially for Scandinavian languages, is a must for a  paper focused on Norwegian. (I assume that the authors are aware of those.)

To start with, the scope of work on datasets for linguistic acceptability is growing, look at a reference list for article on a Russian CoLa dataset: "RuCoLa: Russian Corpus of Linguistic Acceptability". You might need to extend your section on the previous work.

Second, there is a benchmark for Swedish, SuperLim and SuperLim-2, that are inspired by GLUE, see https://spraakbanken.gu.se/en/resources/superlim. This site lists a range of reference articles for datasets and tasks, the major one being :
https://gupea.ub.gu.se/handle/2077/67179. Among the Swedish SuperLim datasets/tasks, there is a Swedish dataset for Linguistic Acceptability, DaLAJ, that is - importantly - also based on learner essays with one-error-per-sentence principles, and was the first of its kind. The authors should acquaint themselves with that dataset and reference it (e.g. https://ep.liu.se/ecp/177/003/ecp2021177003.pdf, https://arxiv.org/abs/2105.06681, https://ecp.ep.liu.se/index.php/sltc/article/view/581)

Third, ScandEval (https://scandeval.github.io/nlu-benchmark/) also lists linguistic acceptability task, and may be worth checking this collection.

Among other comments - it seems that conflation of ASK error-tags ignores some of the tags, e.g. for word order INV and OINV are not named (only O is named), see Appendix.

Results in Table 2 are really impressive. Could it be due to a potential bias in the models to classify sentences as incorrect, since the dataset has only ≈30% of correct sentences with the majority being incorrect ones?

Have you considered correlating the results with proficiency levels represented in the ASK corpus?

Minor comments:

360-362: something is wrong with the language use
371: use "overall performance"
398: add preposition "with"
399-400: fix word order (models NB-BERT's)
402: add preposition "with"


**Paper Type:**

Short paper

---

### Official Review · Reviewer_gK2F · 2023-03-09
**This paper deals with Norwegian datasets appropriate for elaborating the tool evaluating language models' understanding of grammaticality.**

**Rating:** 7
**Confidence:** 3

**Review:**

The authors clearly describe two new datasets for evaluating the understanding language models called the Norwegian corpus of linguistic acceptability (NoCoLA). This is a significant resource for Norwegian and other Scandinavian languages. The main differences between Norwegian and English datasets CoLA and BliMP are mentioned, e.g., NoCoLA consists of naturally produced sentences, instead of controlled and artificially generated ones. The source for both datasets is A Language Learner Corpus of Norwegian as a Second Language (ASK). I think that an error-annotated corpus of native speakers would be also appropriate for this task (if Norwegian has such a corpus).

**Paper Type:**

Short paper

---

### Decision · Program_Chairs · 2023-03-17

Accept